# The Spatial and Temporal Evolution Pattern and Influencing Factors of Urban Human Settlement Resilience in Three Provinces of Northeast China

**Jianjun Liu [1], Xueming Li [1,\*], He Liu [2] and Yishan Song [1]**

[1] School of Geography, Liaoning Normal University, Dalian 116082, China; 18340830644@163.com (J.L.); songyishan0916@163.com (Y.S.)

[2] Department of Geographical Science, Anshan Normal University, Anshan 114007, China; liuhe@mail.asnc.edu.cn

\* Correspondence: lixueming@lnnu.edu.cn

**Abstract:** It is widely recognized that urban resilience is one of the core goals of urban development. As an important part of a city, the resilience level of urban human settlements directly affects the development trend of urban resilience. However, at present, research results on the resilience of urban human settlements are very rare, are mainly concentrated in the central region of China, and rarely take into account the economically backward northeastern region. Therefore, in order to better improve the anti-risk ability of the urban human settlement environment system in three provinces of Northeast China, fully implement the strategic goal of "Comprehensive Revitalization of Northeast China", and achieve high-quality urban development, this paper focuses on 34 prefecture-level cities in three provinces of Northeast China and proposes an urban human settlement resilience evaluation system with 36 indicators in five dimensions, namely, the natural system, human system, housing system, supporting system, and social system. Using the entropy weight method, the Dagum Gini coefficient, and a geographical probe model, the changes in the resilience level of each city from 2005 to 2020 were measured, and the urban living environment was assessed in terms of the adaptability and resilience of the development level in each subsystem based on the temporal and spatial evolution law and its influencing factors. The following conclusions were drawn: (1) The development level of urban human settlement resilience in the three provinces in Northeast China showed an N-shaped development trend from 2005 to 2020, but the regional differences were significant, and the overall spatial pattern was "high in the south and low in the north". (2) In terms of the overall difference, the overall difference in urban human settlement resilience in the three northeastern provinces of China was small: the inter-regional difference was the main source of the difference, and the intra-regional difference was the secondary source. The regional differences were in the order of Heilongjiang Province > Liaoning Province > Jilin Province, indicating that Jilin Province had the smallest difference and that the resilience level of urban human settlements does not show a balanced development trend. In terms of the average Gini coefficient between regions, the order of difference was Liaoning Province–Heilongjiang Province > Jilin Province–Liaoning Province > Jilin Province–Heilongjiang Province, indicating that the difference between Liaoning Province and Heilongjiang Province was the most significant. (3) The "natural system", "human system", "living system", "supporting system", and "social system" had significant spatial and temporal heterogeneity and significantly affected the resilience level of urban human settlements in the three provinces in Northeast China. Among them, the "social system" has always been the main factor affecting the resilience level of urban human settlements.

**Keywords:** resilience of urban human settlements; space–time evolution; influencing factors; three provinces in Northeast China

## 1. Introduction

Due to the rapid expansion of cities and the rapid increase in population, the relationship between people and land in cities has changed, and the urban spatial structure has begun to transform. As a result, urban human settlements have been affected and disturbed by various events such as urbanization, industrialization, and natural disasters [1]; the urban heat island effect [2]; smog [3]; wildfires [4]; and urban waterlogging [5]. The occurrence of a series of disasters has impacted the stability and adaptability of the urban human settlement environment system to a certain extent, seriously slowing down the sustainable development of cities [6]. In this context, understanding the resilience of urban human settlements and planning to address their vulnerabilities are critical to high-quality sustainable urban development, especially in the wake of world health events such as COVID-19, and exploring its spatiotemporal pattern and influencing factors is also of great theoretical significance to enrich the theory of urban human settlement system.

As an important aspect of urban development, urban resilience provides a new perspective for cities to resist disasters and cope with risks. Urban resilience means that when facing the impact of natural or human-caused disasters, a city can not only resist pressure but also adapt to the changing environment on the premise of ensuring its basic functions [7]. This process consists of five stages, namely, resistance, absorption, adaptation, transformation, and recovery [8]. The scientific quantification of urban resilience is an important means for the high-quality construction of resilient cities. Therefore, scholars have engaged in extensive discussions on urban resilience. Considering the research field, the study of resilience has gradually extended from a single natural ecosystem [9] to a complex natural–social system [10,11]. In terms of research content, studies have been mostly conducted from the perspectives of conceptual connotation [12–14], comprehensive evaluation [15,16], influencing factors [17], promotion strategies [18,19], etc. As for research methods, quantitative research initially focused on a single aspect has developed into a combination of qualitative and quantitative research [20], such as comprehensive index methods and measurement methods. In terms of the research scale, it includes countries [21–23], provinces [24], city clusters [25], prefecture-level cities [26], communities [27], and so on. In addition, some scholars have proposed that basic urban green infrastructure (UGI) can improve urban ecological resilience [28] or can be used to study the association between urban form and urban resilience using special methods and exploring the relationship between the two [29]. The abundant research results regarding urban resilience provide theoretical and methodological references for investigating the resilience of urban human settlement systems. On the one hand, a city with a high level of resilience can face sudden disasters without difficulty, while on the other hand, it can quickly adapt to the increasingly complex social and economic environment, thus ensuring the healthy development of the urban system [30]. In recent years, resilient cities have gradually attracted the attention of the Chinese government. In 2020, China's "14th Five-Year Plan" outlined the construction of resilient cities as a national strategy for sustainable development [31]. At the same time, in order to enhance the resilience of Chinese cities, pilot projects such as adaptive cities and sponge cities have been introduced [30]. However, due to the late start of the concept of resilient cities in China, both its theory and practice are not sufficient at present. Therefore, the strategic goal of building resilient cities will take time and effort to achieve.

The governance of urban human settlements is an important problem in the process of urbanization [32]. Generally speaking, urban human settlements involve two types of environments: the hard environment and the soft environment. The hard environment emphasizes the unity of nature, humanity, and space, which mainly includes living conditions, environmental quality, and public facilities. The soft environment focuses on the sum of all non-material features, such as life comfort, social order, security, and a sense of belonging [33]. Research on urban human settlements mainly originates from the theory of "human settlement" put forward by the Greek scholar Doxiadis [34] and the theory of "human settlement science" put forward by the academician Wu Liangyong [35]. Scholars mostly start with an evaluation of the human settlement environment [36] and focus on

evaluation indicators of the urban human settlement environment based on the five systems of the human settlement environment, nature, society, habitation, and support [37]. The spatial and temporal evolution pattern [38], influencing factors [39], and driving mechanism [40], as well as the interactions among human daily activities, the residential spatial structure, and the physical geographical environment [41] are explored by investigating regional characteristic indicators. At the same time, some scholars have begun to pay attention to the vulnerability of urban human settlement systems, and the research results provide theoretical support for an understanding of the system's response to risks and an improvement in its ability to reduce vulnerability [42]. As an inherent attribute of the system, resilience aims to maintain the stability of system operation and improve the sustainable development capacity of the system [1]. Recently, Li Xueming conducted a preliminary exploration of the resilience of human settlements in the Yangtze River Delta urban agglomeration based on the DPSIR model [43]; Peng Kunjie discussed in more detail the spatial and temporal evolution characteristics of urban human settlement resilience in the Yangtze River Delta region [44]; and Zhou Xiaoqi measured and evaluated the resilience of urban human settlements in China as a whole and built a driving mechanism based on geographic detectors [45].

To summarize, although the current research results on resilience and urban human settlements are quite abundant, there are still several deficiencies. One is that "resilience" mainly depends on the attributes and characteristics required by the carrier, which remain independent when applied to different disciplines. As a complex and large system, urban human settlements have strong openness and inclusiveness, which are closely related to the research perspective of resilience. In view of the significant vulnerability of human life and the urban economy, it is necessary to introduce the theory of resilience into the human settlement system, aiming to enrich the theoretical framework of urban human settlements [46]. Second, the current research on urban resilience mainly constructs an index system from the four dimensions of economy, society, ecology, and engineering, and the index focuses more on external environmental factors, ignoring the internal connection and mechanism interactions among human settlements, especially the subjective dynamic role of the "human system" [47]. Third, the current research on urban human settlements mostly starts from a comprehensive evaluation and influencing factors, and there is a lack of research on how to deal with risks and resist shocks in the human settlement system. Moreover, there are still few studies that measure its space–time evolution from both time and space perspectives. Therefore, it is necessary and relevant to combine the theory of resilience with the urban human settlement system. In the context of climate change and rapid urbanization, improving the resilience of urban human settlements is conducive to strengthening urban emergency management capabilities and is an important way to achieve the high-quality sustainable development of cities.

As one of the four major economic sectors in China [48], the three provinces in Northeast China represent not only the largest heavy industry base in China but also an important grain production base. However, over time, a series of problems such as environmental pollution, resource depletion, industrial structure imbalance, and market environment change have led to the phenomena of production lags, enterprise closures, economic regression, and brain drain in the three provinces in Northeast China. By studying the resilience level of urban human settlements in the three provinces in Northeast China, we can effectively improve the resistance, adaptability, and resilience of the three eastern provinces in the face of sudden disasters. In summary, 34 prefecture-level cities in the three provinces in Northeast China were selected as research areas, and methods such as the entropy weight method, the Dagum Gini coefficient, and the geographical detector were used to diagnose and perform empirical research on the resilience of urban human settlements in order to provide a new perspective for the study of urban human settlements and a reference for the revitalization of Northeast China, the new urbanization of Northeast China, and the implementation of territorial spatial planning.

## 2. Overview of the Study Area and Data Sources

### 2.1. Overview of the Study Area

Northeast China is one of the four economic sectors in China. Due to its geographical location and the restriction of the revitalization of Northeast China, the vulnerability of the urban human settlement environment has become an important obstacle in promoting high-quality regional development and realizing the revitalization of Northeast China [49]. The details are as follows. (1) By the end of 2022, the permanent population of the three provinces in Northeast China was approximately 96.44 million. It had decreased by nearly 10 million compared with 2011. The urban population loss was serious, and the population structure was unbalanced. The shortage of talent has led to the inferior position of the northeastern cities in the field of scientific and technological innovation, and they cannot keep up with the pace of modern development. (2) By the end of 2022, the GDP of Northeast China accounted for only 5% of the country's total, and the per capita disposable income of the cities was also lower than the national average, indicating that the overall economic level of Northeast China was low and the industrial structure's contradictions were prominent. (3) The three provinces in Northeast China are located in the "chicken head" of the map of China, and long-term transportation relies solely on the Liaoxi Corridor, the Beijing–Harbin Expressway, and the only port, which is in Dalian. The transportation infrastructure needs to be improved, as well as the urban governance capacity. This series of problems not only reduced the happiness and satisfaction of residents, but it also seriously affected the healthy and sustainable development of the urban human settlements. Thus, the ability of the urban human settlements to withstand risks needs to be considered. Therefore, 34 prefecture-level cities in the three provinces in Northeast China are selected as the main research areas in this paper (Figure 1). Daxinganling in Heilongjiang Province and Yanbian in Jilin Province are not included in this research due to a lack of data. This study aims to enhance the resilience of urban human settlements and promote high-quality development in the three provinces in Northeast China based on the strategic background of the revitalization of the Northeast.

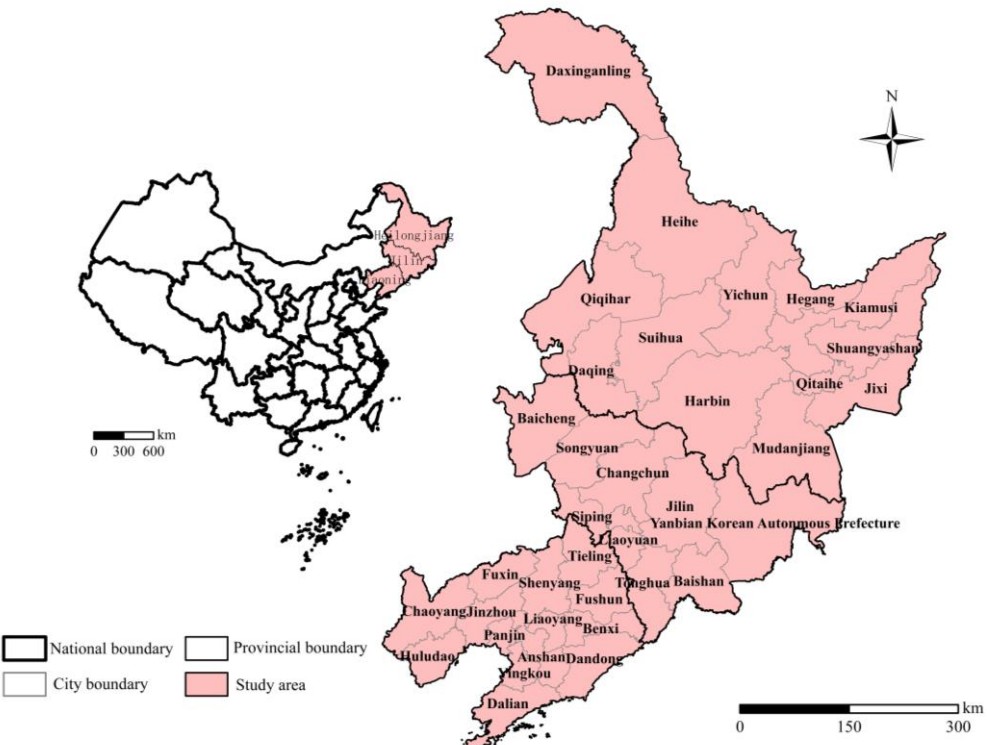

**Figure 1.** Administrative zoning map of 34 prefecture-level cities in three eastern provinces.

## 2.2. Data Sources

There are two types of data needed in this study (Table 1). One is the raster data that express the natural elements of the region, including the 30 m resolution digital elevation model (DEM) and 500 m resolution NDVI data from the Data Center of Resources and Environmental Sciences, Chinese Academy of Sciences. Via the relief degree, we used the differential ArcGIS10.2 software (Version number: 10.2.0.3384) to perform DEM data calculations. Meteorological data such as the annual mean temperature and annual mean precipitation were obtained from the official websites of municipal meteorological bureaus and the National Greenhouse Data System. Second, statistical data representing the population and economic and social development were obtained. Among them, data on the natural population growth rate, population density, urban domestic sewage treatment rate, harmless treatment rate of household garbage, comprehensive utilization rate of general industrial solid waste, green coverage rate of built-up areas, per capita GDP, per capita disposable income of urban residents, and average wage of working employees were directly derived from the China Urban Statistics Yearbooks for 2006, 2011, 2016, and 2021 and provincial and municipal statistical yearbooks, as well as provincial and municipal statistical bulletins for 2005, 2010, 2015, and 2020. Some of the missing data were completed using Excel software (Version number: 2311 Build 16.0.17029.20028) with the interpolation method, and the rest of the index data were obtained after the calculation of the statistical data.

**Table 1.** Sources of research data.

| Data Name | Year | Data Description | Data Source |
|---|---|---|---|
| Degree of relief, DEM | 2010 | 30 m spatial resolution digital elevation model | http://www.gscloud.cn/(accessed on 1 January 2023) |
| NDVI | 2005, 2010, 2015, 2020 | 500 m/16 d, normalized vegetation index monthly annual mean data | http://www.resdc.cn/( accessed on 15 January 2023) |
| Meteorological station data | 2006, 2011, 2016, 2021 | Annual mean data from each meteorological station, including precipitation, temperature, humidity and other data | Htttp://data.cma.cn/ (accessed on 5 February 2023) |
| Statistical yearbook data | 2006, 2011, 2016, 2021 | Contains population, land, economic and other statistics | "China City Statistical Yearbook" and provincial and municipal statistical Yearbook |

## 3. Research Methods

### 3.1. Construction of the Resilience Index System of Urban Human Settlements

The urban human settlement environment system is a complex and large system that mainly reveals the interaction between humans, cities, and nature. The Chinese academician Wu Liangyong constructed the basic framework of human settlement environment science for the first time, dividing the urban human settlement system into five subsystems: the natural system, human system, social system, residential system, and supporting system [35]. Based on this theoretical framework, the theoretical connotations of urban resilience and the urban human settlement system are integrated, and the research achievements of Li Xueming [43], Peng Kunjie [44], and other researchers are combined. This study suggests that the resilience of urban human settlements is subordinate to urban resilience in general. Urban resilience focuses on social indicators; it mainly reflects the response of the urban public service system in the face of natural disasters, and it often ignores the internal connections and mechanisms of the human settlement system. The resilience of urban human settlements is more strongly related to human factors, emphasizing "people" as the main body, and is manifested in the qualities of anti-risk and sustainability. The internal structure, factors, and scale of the five subsystems of human settlement gradually change from a low level to a high level when responding to internal and external shocks. Therefore, following the principles of scientificity, systematicity, independence, and data availability, this study divides the evaluation index system for the resilience level of urban human settlements into 5 criterion layers and 36 indicator layers (Table 2). Among them, in

order to combine the regional characteristics of rich natural resources and developed heavy industry in the three provinces in Northeast China, in particular, indicators with regional characteristics, such as the vegetation index [50] and $SO_2$ emissions per unit of industrial output [51], are added to the index system.

**Table 2.** Evaluation index system of resilience level of urban human settlements.

| Target Layer | Criterion Layer | Element Layer | Index Layer | Indicator Attributes |
|---|---|---|---|---|
| Resilience of urban human settlements | Natural system resilience | Topographic flatness | Relief of relief ($X_1$) | - |
| | | | DEM elevation ($X_2$) | - |
| | | Basic climatic condition | Annual mean temperature ($X_3$) | * |
| | | Natural water supply capacity | Mean annual precipitation ($X_4$) | * |
| | | Green environment | Vegetation index ($X_5$) | + |
| | Human system resilience | Population structure | Proportion of urban population ($X_6$) | + |
| | | | Gender balance rate ($X_7$) | - |
| | | Population growth trend | Natural rate of population growth ($X_8$) | - |
| | | Sparsity | Population density ($X_9$) | - |
| | | Employment structure | The proportion of employees in the tertiary industry ($X_{10}$) | + |
| | | Unemployment structure | The proportion of people who are unemployed ($X_{11}$) | - |
| | Dwelling system resilience | Gas resource supply capacity | Per capita gas consumption ($X_{12}$) | + |
| | | Environmental conservation level | Per capita green space ($X_{13}$) | + |
| | | Water supply capacity | Per capita daily domestic water consumption ($X_{14}$) | + |
| | | Educational condition | Number of students in regular institutions of higher learning per 10,000 ($X_{15}$) | + |
| | | Internet popularity | Internet per 10,000 people ($X_{16}$) | + |
| | | Electric power development level | Per capita domestic electricity consumption ($X_{17}$) | - |
| | Support system resilience | Traffic facility level | Per capita urban road area ($X_{18}$) | + |
| | | | There are buses for every 10,000 people ($X_{19}$) | + |
| | | Medical condition | Number of hospital beds per 10,000 people ($X_{20}$) | + |
| | | Sewage treatment level | Urban domestic sewage treatment rate ($X_{21}$) | + |
| | | Refuse disposal level | Harmless treatment rate of household garbage ($X_{22}$) | + |
| | | Comprehensive utilization of waste | Comprehensive utilization rate of general industrial solid waste ($X_{23}$) | + |
| | | Urban greening level | Green coverage rate of built-up area ($X_{24}$) | + |
| | | Urban construction level | The proportion of urban construction land in urban area ($X_{25}$) | + |
| | | Environmental pressure | $SO_2$ emissions per unit of industrial output ($X_{26}$) | - |
| | | Public resource guarantee | Library volume per 100 people ($X_{27}$) | + |
| | Social system resilience | Economic aggregate | Per capita GDP ($X_{28}$) | + |
| | | Social management level | Proportion of personnel in public administration and social organizations ($X_{29}$) | + |
| | | Scientific and technological input | Proportion of investment in science and technology ($X_{30}$) | + |
| | | Educational input | Proportion of investment in education ($X_{31}$) | + |
| | | Economic resilience | Urban per capita disposable income ($X_{32}$) | + |
| | | Level of national economic development | Per capita retail sales of consumer goods ($X_{33}$) | + |
| | | Development level of posts and telecommunications | Postal service per capita ($X_{34}$) | + |
| | | | Per capita telecommunications service ($X_{35}$) | + |
| | | Resident wage income | Average wage of working staff ($X_{36}$) | + |

Note: "+" represents a positive indicator; "-" indicates a negative indicator; "*" indicates a moderate indicator; and the average value is taken as the most moderate number [52].

### 3.2. Entropy Weight Method

The entropy method originated from fragrant information theory and is used to reveal the uncertainty in information source signals [53]. The methods used to determine weight indicators mainly include subjective and objective methods. In this study, the relatively objective entropy method was selected for weighting, and a time variable was added to determine the weights of the indicators, mainly according to the degree of dispersion of the selected indicators, which can avoid the influence of subjective factors to a certain extent [54]. It can also quantify some factors with blurred boundaries that are difficult to quantify. If the information entropy of an indicator is large, it means that its variability

is smaller, the information that it contains is reduced, and its weight is smaller, and vice versa [55].

(1)　Before using the entropy method to calculate the weight of the index, first, standardize the data:

$$\text{Positive indicators}: \ X_{ij} = \frac{X - \min(X_i)}{\max(X_i) - \min(X_i)} \tag{1}$$

$$\text{Negative indicator}: \ X_{ij} = \frac{\max(X_i) - X}{\max(X_i) - \min(X_i)} \tag{2}$$

Moderate indicators:

$$X_{ij} = \left( y_{max} - \left| X - \sum_{n}^{1} X_i \right| \right) / \left( y_{max} - y_{\min} \right) \tag{3}$$

where $X$ is the original value; $X_{ij}$ is the result of index standardization; $y_{min}$ is the minimum value of the interpolation between the original index value and the mean value; and $y_{max}$ is the maximum value of the interpolation between the original index value and the mean value.

(2)　Calculate the contribution rate of indicators:

$$S_{ij = \frac{X_{ij}}{\sum_{i=1}^{n} X_{ij}}} \tag{4}$$

where $S_{ij}$ is the contribution rate of item $j$ to the index in $i$ and n is the number of years.

(3)　Calculate index entropy:

$$P_j = k \sum_{i=1}^{m} S_{ij} \ln \left( S_{ij} \right), k = 1/\ln(m) \tag{5}$$

where $P_j$ is the entropy value of the $j$ index; m is the number of indicators; and n indicates the number of years.

(4)　Calculate index redundancy:

$$Z_j = 1 - P_j \tag{6}$$

where $Z_j$ is the redundancy of item $j$.

(5)　Calculate index weights:

$$W_j = \frac{Z_j}{\sum_{1}^{n} Z_j} \tag{7}$$

(6)　Calculate the urban human settlements resilience score:

$$E_j = \sum_{i=1}^{m} Z_j W_j \tag{8}$$

where $E_j$ is the resilience score of an urban human settlement.

### 3.3. Dagum Gini Coefficient Decomposition Method

The methods used to measure inter-regional differences mainly include the Theil index and traditional Gini coefficient. However, these methods have certain limitations and cannot fully consider the spatial distribution of subsamples [56]. The Daugm Gini coefficient fully compensates for these deficiencies and calculates not only regional global differences but also intra-regional differences, interval differences, and the super-variable density [57]. Therefore, the Dagum Gini coefficient decomposition method was used in this study to measure the differences in the development levels of urban human settlements'

resilience in the three northeastern provinces. The overall Gini coefficient is calculated as follows:

$$G = \frac{1}{2n^2\mu}\sum_{i=1}^{k}\sum_{j=1}^{k}\sum_{h=1}^{n_i}\sum_{r=1}^{n_j}|y_{ih} - y_{jr}| \tag{9}$$

where $G$ is the overall Gini coefficient of the resilience of urban human settlements in the study area; $k$ is the number of subgroups, including Heilongjiang, Jilin, and Liaoning; $n$ is the number of cities in the three provinces in Northeast China; $n_i(n_j)$ is the number of cities in a subgroup of $i(j)$; $\mu$ is the average of the composite index of the resilience of all urban human settlements; and $y_{ih}(y_{jr})$ is a composite index of the resilience of any city $i(j)$ in the three eastern provinces. The smaller the $G$ value, the more balanced the development of human settlements' resilience between cities, and vice versa [58].

In addition to analyzing the difference degree of the development of the toughness level of the overall urban human settlement, this paper also constructed the Gini coefficient $G_{jj}$ of the toughness level of the human settlements of subgroup $j$ and the Gini coefficient $G_{ih}$ between subgroups $j$ and $h$, whose calculation formulas are as follows:

$$G_{jj} = \left(\sum_{i=1}^{n_j}\sum_{r=1}^{n_j}|y_{ji} - y_{jr}|\right)/2n_j^2\mu_j \tag{10}$$

$$G_{ih} = \left(\sum_{i=1}^{n_j}\sum_{r=1}^{n_h}|y_{ji} - y_{hr}|\right)/n_jn_h(\mu_j + \mu_h) \tag{11}$$

where $\mu_j$ and $\mu_h$ are the average toughness levels of subgroups $j$ and $h$, respectively, and the other indexes are shown above. In order to further construct the Dagum Gini coefficient subgroup decomposition function, the following variables are further defined:

$$D_{jh} = \frac{d_{jh} - p_{jh}}{d_{jh} + p_{jh}} \tag{12}$$

$$d_{jh} = \int_0^\infty dF_j(y)\int_0^y (y - x)dF_h(x) \tag{13}$$

$$p_{jh} = \int_0^\infty dF_h(y)\int_0^y (y - x)dF_j(x) \tag{14}$$

$$P_j = \frac{n_j}{n} \tag{15}$$

$$S_j = n_j\mu_j/n\mu \tag{16}$$

where $D_{jh}$ represents the mutual influence of toughness level development between subgroups $j$ and $h$; $d_{jh}$ and $p_{jh}$ represent the mathematical expectation of $y_{ji}\text{-}y_{hr} > 0$ and $y_{ji}\text{-}y_{hr} < 0$ in subgroups $j$ and $h$, respectively; and $F$ represents the cumulative density function of the subgroup's resilience level.

In this study, the overall Gini coefficient $G$ of Dagum was decomposed into $Gw$, $G_{nb}$, and $Gt$, and the relationship between them satisfied $G = G_w + G_{nb} + G_t$. The calculation formulas are as follows:

$$G_w = \sum_{i=1}^{n_j}G_{jj}P_jS_j \tag{17}$$

$$G_{nb} = \sum_{j=2}^{k}\sum_{h=1}^{j-1}G_{jh}(P_jS_h + P_hS_j)D_{jh} \tag{18}$$

$$G_t = \sum_{j=1}^{k}\sum_{h=1}^{j-1}G_{jh}(P_jS_h + P_hS_j)(1 - D_{jh}) \tag{19}$$

### 3.4. Geographic Detector

The geographical detector mainly determines the interaction characteristics between two variables by comparing the $q$ values of single- or two-factor interactions [59]. This

tool is not limited to traditional statistical methods, and it can more objectively show the degree of explanation between the independent variable and the dependent variable with the non-linear hypothesis. Therefore, this study used the geographical detector to explore the main influencing factors of urban human settlement resilience in the three eastern provinces [60]. Firstly, the indicator data were imported into ArcGIS10.2 software and were then discretized using the paragraphpoint method to convert them into type variables. Then, based on the geographic detector model, the influence of each index on the resilience level of urban human settlements was calculated. The calculation formula is as follows:

$$q = 1 - \frac{\Sigma_{h=1}^{L} N_h \sigma_h^2}{N \sigma^2} = 1 - \frac{SSW}{SST} \tag{20}$$

$$SSW = \Sigma_{h=1}^{L} N_h \sigma_h^2, \quad SST = N \sigma^2 \tag{21}$$

where $q$ represents the explanatory power of the factor; the range is [0, 1], and the more closely it approaches 0, the smaller the explanatory power is. $h$ represents the stratification of explanatory variables or explained variables; $N_h$ is the number of layer $h$ units; $N$ represents the number of units in the whole area; $\sigma^2$ represents the variance in the $Y$ value across the region; $\sigma_h$ represents the variance in layer $h$; $SSW$ represents the sum of the in-layer variances; and $SST$ represents the sum of the variances in the whole region.

## 4. Result Analysis

### 4.1. Overall Change Characteristics

The resilience level of urban human settlements was calculated based on Formulas (1)–(8). In order to describe the change trend in the resilience level of urban human settlements more clearly, Figure 2 is presented, which was obtained using Excel software. From 2005 to 2020, the average development level of urban human settlement resilience in the three provinces in Northeast China showed an "N"-type trend, and the distribution range was [0.2513, 0.4530] (Figure 2). During this period, 2010 and 2015 were the turning point years, and the lowest value was reached in 2015. The main reasons are as follows. In 2005, the most critical year for China's reform, China proposed policies conducive to urban planning and improving residents' happiness from the perspectives of land management, resident consumption, infrastructure, economic reform, the housing system, landscape beautification, animal and plant protection, etc., which improved the resilience levels of urban human settlements. After the 17th National Congress of the Communist Party of China, the integration of urban and rural areas was formally proposed. Moreover, a series of natural disasters occurred in 2012, including a tsunami, earthquake, hail, flash floods, mudslides, etc. To a certain extent, these seriously hindered the growth of the resilience level of the urban human settlements. After this, China began to adopt the construction of a "smart city" and "sponge city" as a key goal, actively optimize urban public services, focus on highlighting the primary position of the city, and greatly promote the development of the resilience level of the urban living environment.

### 4.2. Characteristics of Spatiotemporal Pattern Evolution

(1) Based on the data characteristics of the urban human settlement resilience evaluation index, the natural break point method of ArcGIS10.2 was used to divide the urban human settlement resilience evaluation indices of the three northeastern provinces of China in 2005, 2010, 2015, and 2020 into five grades: low level, low level, medium level, high level, and high level (Figure 3). Overall, the resilience level of urban human settlements in the three provinces in Northeast China presented a spatial pattern of "high in the south and low in the north", which is mainly manifested as "stable growth in the southwest and a gradual decline in the north". The specific characteristics are as follows. The middle- and high-level regions decreased, and the dispersion of the high-level regions was significant, showing a "multi-core" divergence trend. The dispersion characteristics of the higher-level regions were weakened, and most of

them were gathered in the central and southern regions. The characteristics of middle-level regional agglomeration were obvious and concentrated in the southern part of the three provinces in Northeast China. In 2005, a high level of urban human settlement resilience was noted in Daqing, Shenyang, and Dalian; the high level showed a "double core" pattern that was distributed in Harbin, Panjin, Mudanjiang, Benxi, Anshan, Liaoyang, Dandong, and Fushun. A medium level was noted in Jimusi, Changchun, Jilin, Liaoyuan, Baishan, Tonghua, Yingkou, and Jinzhou. In 2010, three cities including Daqing, Shenyang, and Dalian still presented a high level; six cities including Harbin, Changchun, Benxi, Fushun, Anshan, and Panjin showed a high level; and seven cities including Mudanjiang, Jilin, Liaoyuan, Tonghua, Dandong, Yingkou, and Jinzhou had a medium level. In 2015, there were only two cities with a high level: Shenyang and Dalian, and the number of higher-level regions also decreased, namely, Daqing, Baicheng, Harbin, Changchun, and Benxi, while the middle level was mainly distributed in 14 cities in the east and south including Jilin, Liaoyuan, Anshan, Liaoyang, and Dandong. In 2020, Shenyang and Dalian were still at a high level of resilience, and a higher level was seen in nine cities: Changchun, Fuxin, Jinzhou, Panjin, Yingkou, Anshan, Benxi, Fushun, and Liaoyang. The medium level was further concentrated in the southwest direction of the three provinces in Northeast China, including Daqing, Harbin, Jilin, Tieling, Dandong, Chaoyang, and Huludao.

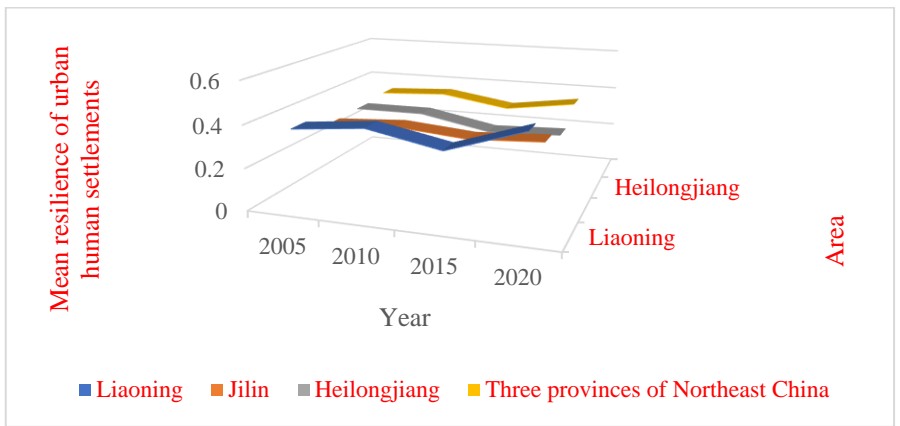

**Figure 2.** Mean change in the resilience level of urban human settlements in three provinces in Northeast China.

(2) A low level and low-level areas tended to increase, and clusters were seen in the northern part of the three provinces in Northeast China. Specifically, in 2005, a low level was noted in Suihua, Baicheng, Chaoyang, and Qitaihe, totaling four cities, and a low level was noted in Heihe, Qiqihar, Yichun, Hegang, Shuangyashan, and another 14 cities. In 2010, the number of low-level areas increased to 10 cities, including Heihe, Qiqihar, Shuangyashan, Huludao, and Songyuan, etc., and a low level was seen in Yichun, Hegang, Jiamusi, Jixi, Baishan, Tieling, Fuxin, and Liaoyang, totaling eight cities. In 2015, the lower-level cities were Heilongjiang, Qiqihar, Suihua, Songyuan, Siping, and Jixi, while the lower-level cities included Yichun, Hegang, Jiamusi, Qitaihe, Mudanjiang, Tieling, and Chaoyang. In 2020, the number of low- and low-level cities, including Heihe, Qiqihar, Suihua, Hegang, Jimusi, and another 16 cities, increased compared with 2020, which shows, to a certain extent, that the development level of urban human settlement resilience in the three provinces in Northeast China worsened from 2011 to 2020.

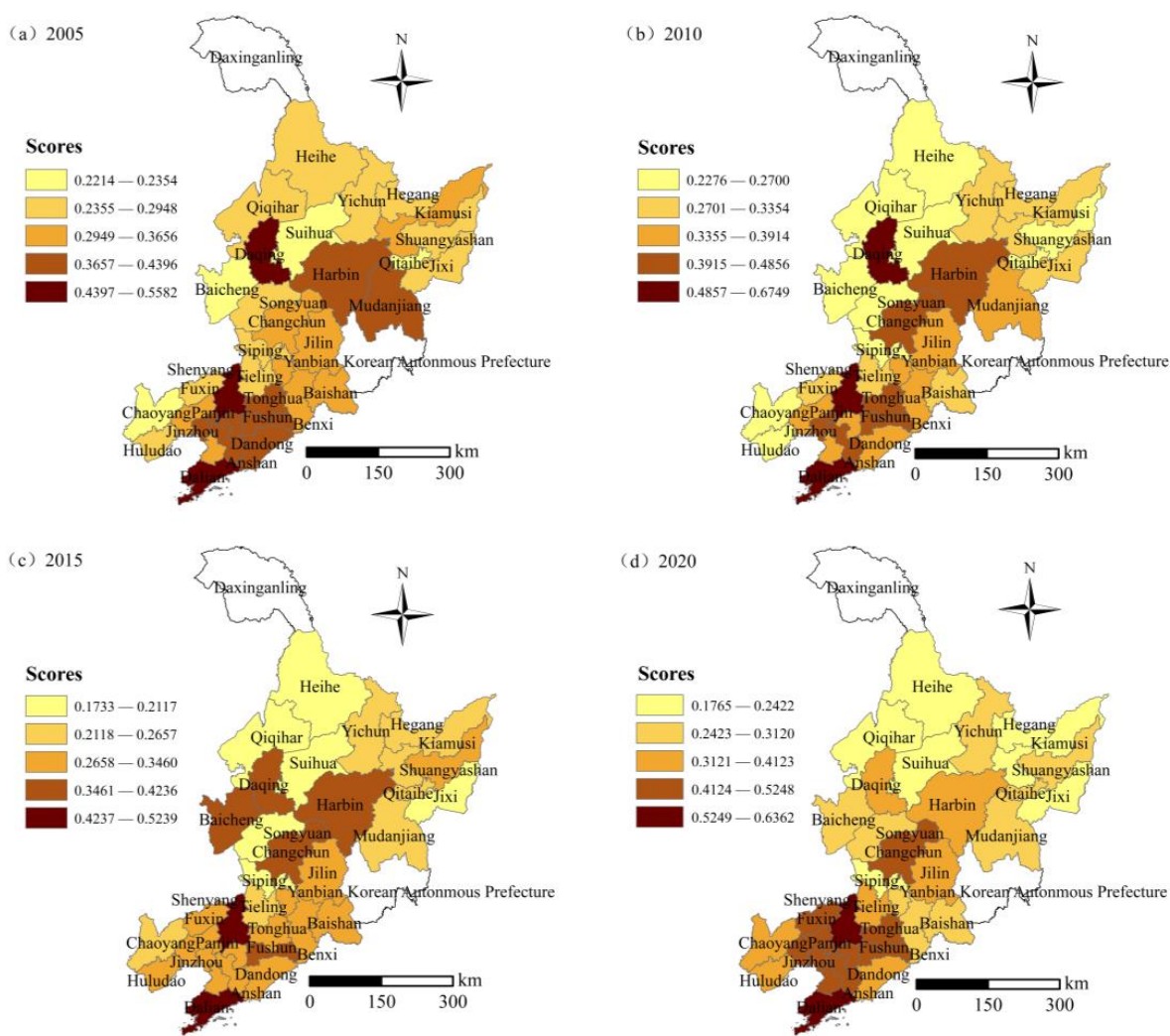

**Figure 3.** Spatial and temporal variation in urban human settlement resilience of three provinces of Northeast China.

### 4.3. Analysis of the Difference in the Resilience Level of Urban Human Settlements among the Three Provinces

#### 4.3.1. Overall Difference Analysis

In this study, the Dagum Gini coefficient was used to analyze the overall differences in the development level of urban human settlement resilience in the three provinces in Northeast China. Table 3 is based on Formulas (9)–(19), and it can be seen that (1) from the perspective of overall differences, there is a small overall gap in the resilience of urban human settlements in the three provinces in Northeast China. The Gini coefficient is between 0.139 and 0.179, showing an N-shaped development trend from 2005 to 2020. Specifically, during the first stage (2005–2010), the resilience of urban human settlements showed a rapidly rising trend; in the second stage (2010–2015), the resilience level tended to decline slightly; in the third stage (2015–2020), the resilience level of urban human settlements reached a peak of 0.179, indicating that the imbalance in the resilience level of urban human settlements in the three provinces in Northeast China has always existed and has become increasingly significant. (2) From 2005 to 2020, regarding the average development level of urban human settlement resilience in the three provinces in Northeast China, regional differences accounted for 46.78% of the total differences, while intra-regional differences accounted for only 28.73%. (3) The contribution rate of intra-regional differences showed a decreasing trend year by year, from 33.16% in 2005 to 21.07% in 2020.

(4) The inter-regional contribution rate showed an upward trend in general, increasing rapidly from 31.47% to 70.45%.

**Table 3.** Differences in the contribution rate of urban human settlement resilience development in the three provinces in Northeast China.

| Year | Population Gini Coefficient | Gini Coefficient in the Region | Interregional Gini Coefficient | Supervariable Density | Intra-Regional Contribution Rate | Interregional Contribution Rate |
|---|---|---|---|---|---|---|
| 2005 | 0.139 | 0.046 | 0.044 | 0.049 | 33.16% | 31.47% |
| 2010 | 0.165 | 0.052 | 0.06 | 0.052 | 31.83% | 36.40% |
| 2015 | 0.154 | 0.047 | 0.068 | 0.039 | 30.21% | 44.23% |
| 2020 | 0.179 | 0.038 | 0.126 | 0.015 | 21.07% | 70.45% |

### 4.3.2. Intra-Regional Difference Analysis

Formula (10) was used to calculate the Gini coefficient of urban human settlement resilience in the three eastern provinces during 2005, 2010, 2015, and 2020 (Table 4), and the following conclusions were drawn. (1) From the mean value, the intra-regional Gini coefficients of urban human settlement resilience in the three provinces in Northeast China during the 2005–2020 period were as follows: Heilongjiang > Liaoning > Jilin Province. Compared with other regions, the uneven level of resilience of the urban human settlements was the most serious here, and it hindered the modernization process of the three provinces in Northeast China and the implementation of the revitalization strategy policy of the Northeast to a certain extent. (2) From 2005 to 2020, the intra-regional differences in Jilin Province generally showed an expanding trend from 0.083 to 0.125. Among them, the period from 2005 to 2010 showed rapid growth, and the development trend from 2010 to 2020 was stable. (3) From 2005 to 2020, although the intra-regional differences in both Liaoning Province and Heilongjiang Province showed a trend of "first increasing and then decreasing", the intra-regional differences in Liaoning Province gradually narrowed from 0.144 in 2005 to 0.90 in 2020, indicating that the unbalanced development of urban human settlement resilience in Liaoning Province was alleviated, while Heilongjiang Province showed the opposite.

**Table 4.** Intra-regional difference coefficient of urban human settlement resilience development level in the three provinces in Northeast China from 2005 to 2020.

| Year | Jilin | Liaoning | Heilongjiang |
|---|---|---|---|
| 2005 | 0.083 | 0.144 | 0.129 |
| 2010 | 0.126 | 0.154 | 0.147 |
| 2015 | 0.125 | 0.132 | 0.137 |
| 2020 | 0.125 | 0.090 | 0.130 |
| Average | 0.115 | 0.130 | 0.136 |

### 4.3.3. Difference Analysis among the "Three Provinces"

Formula (11) was used to calculate the inter-provincial difference data for the three eastern provinces. In order to express the results more clearly, Excel software was used to obtain Figure 4, to describe the evolution trend in the inter-regional differences in the resilience levels of urban human settlements in the three provinces of Northeast China. The following conclusions were drawn. (1) In terms of the mean value of the inter-regional Gini coefficient, in descending order, they were Liaoning Province—Heilongjiang Province, Jilin Province—Liaoning Province, Jilin Province—Heilongjiang Province. (2) Generally speaking, the differences between Liaoning Province and Heilongjiang Province, and between Jilin Province and Jilin Province, were expanding, and the Gini coefficient rose from 0.159, 0.144, and 0.117 in 2005 to 0.266, 0.201, and 0.155 in 2020, respectively. (3) The difference between Liaoning Province and Heilongjiang Province was the largest. Specifically, the

difference Gini coefficient between Jilin Province and Heilongjiang Province showed a "√" development trend, and 2015 was the turning point year. The evolution trend in the regional difference level between Liaoning and Heilongjiang and Jilin and Liaoning was of the "N" type, and it was divided into three stages: the first stage (2005–2010), the second stage (2010–2015), and the third stage (2015–2020). Their development trends were, respectively, growth—decline—rapid growth. This shows that the difference in the resilience level among the regions in the three provinces in Northeast China decreased during the 2010–2015 period, and the differentiation phenomenon tended to converge, but the opposite was true in other years.

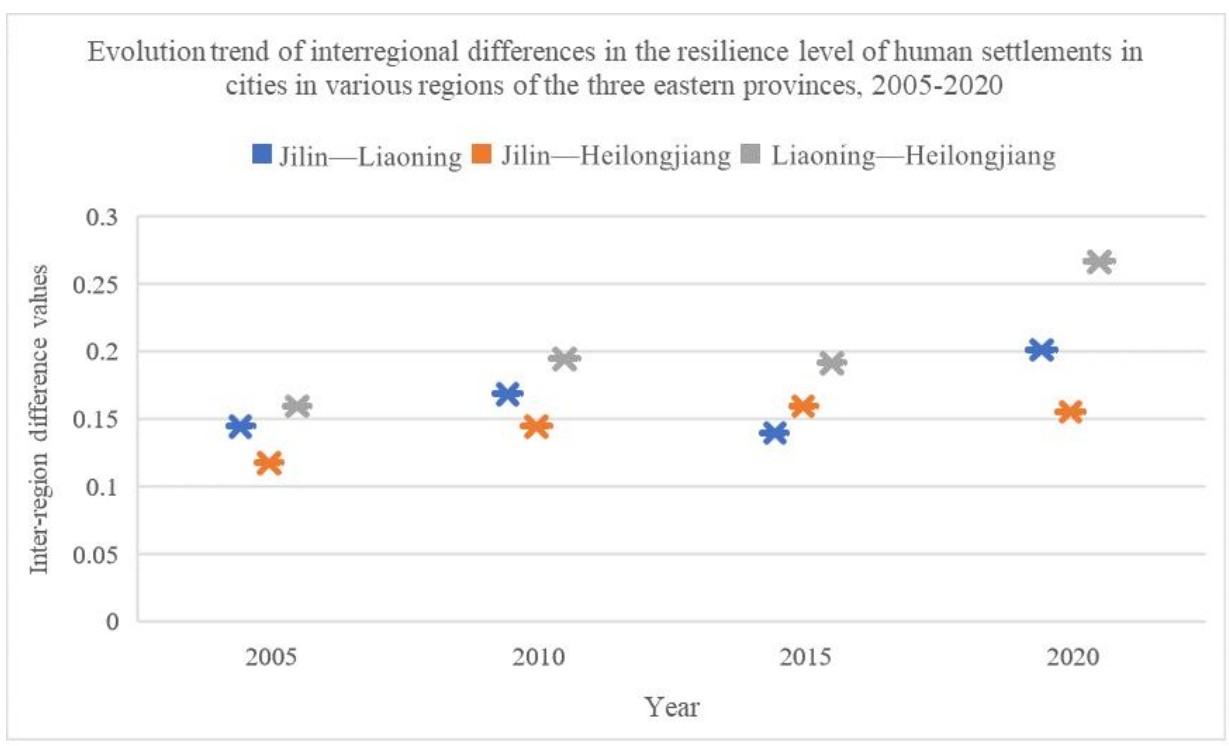

**Figure 4.** Evolution trend in the inter-regional differences in the resilience level of urban human settlements in the three provinces in Northeast China from 2005 to 2020.

*4.4. Analysis of Influencing Factors on the Resilience Level of Urban Human Settlements*

4.4.1. Selection of Influencing Factors

Based on the spatiotemporal evolution characteristics of urban human settlement resilience in the three provinces in Northeast China, the main and secondary factors of the spatiotemporal differentiation of urban human settlement resilience are further discussed and analyzed. First, a correlation analysis was used to judge the effect of indicators on the resilience of urban human settlements. If the correlation coefficient is positive, it indicates that the indicators promote an improvement in the resilience of urban human settlements, and vice versa. According to Table 5, in addition to the natural population growth rate ($X_8$), per capita GDP ($X_{23}$), and per capita urban disposable income ($X_{32}$), other indicators have a significant impact on the resilience level of urban human settlements. Among them, the vegetation index ($X_5$), proportion of employees in the tertiary industry ($X_{10}$), per capita gas consumption ($X_{12}$), per capita postal service ($X_{34}$), and average wage of employees ($X_{36}$) were significantly correlated at the 0.05 level, and other indicators were significantly correlated at the 0.01 level.

**Table 5.** Pearson correlation analysis results.

| Index | $X_1$ | $X_2$ | $X_3$ | $X_4$ | $X_5$ | $X_6$ | $X_7$ | $X_8$ | $X_9$ | $X_{10}$ | $X_{11}$ | $X_{12}$ |
|---|---|---|---|---|---|---|---|---|---|---|---|---|
| Correlation coefficient | −0.272 ** | −0.194 * | 0.322 ** | 0.135 ** | −0.158 * | 0.704 ** | −0.323 ** | −0.091 | 0.592 ** | 0.062 * | −0.121 * | −0.117 * |
| Significant level | 0.000 | 0.043 | 0.000 | 0.000 | 0.017 | 0.000 | 0.000 | 0.769 | 0.000 | 0.042 | 0.000 | 0.045 |
| **Index** | $X_{13}$ | $X_{14}$ | $X_{15}$ | $X_{16}$ | $X_{17}$ | $X_{18}$ | $X_{19}$ | $X_{20}$ | $X_{21}$ | $X_{22}$ | $X_{23}$ | $X_{24}$ |
| Correlation coefficient | 0.692 ** | 0.228 ** | 0.367 ** | 0.555 ** | 0.469 ** | 0.278 ** | 0.557 ** | 0.232 ** | 0.17 ** | 0.257 ** | −0.008 | 0.324 ** |
| Significant level | 0.000 | 0.000 | 0.000 | 0.000 | 0.000 | 0.000 | 0.000 | 0.000 | 0.000 | 0.000 | 0.939 | 0.000 |
| **Index** | $X_{25}$ | $X_{26}$ | $X_{27}$ | $X_{28}$ | $X_{29}$ | $X_{30}$ | $X_{31}$ | $X_{32}$ | $X_{33}$ | $X_{34}$ | $X_{35}$ | $X_{36}$ |
| Correlation coefficient | 0.145 ** | 0.104 ** | 0.715 ** | 0.204 ** | −0.384 ** | 0.404 ** | 0.232 ** | 0.112 | 0.459 ** | 0.078 * | 0.640 ** | 0.160 * |
| Significant level | 0.000 | 0.000 | 0.000 | 0.000 | 0.000 | 0.000 | 0.000 | 0.268 | 0.000 | 0.044 | 0.000 | 0.013 |

Note: The meanings of $X_1$ to $X_{36}$ are the same as those in Table 2. ** indicates that the indicators are significantly correlated at the 0.01 level, and * indicates that the indicators are significantly correlated at the 0.05 level.

The urban human settlement environment is a complex and huge comprehensive system, and its resilience level is determined by each subsystem. Therefore, in order to further explore the internal mechanism of the spatiotemporal evolution pattern of the urban human settlement resilience development level in the three provinces in Northeast China, this study focused on the main influencing factors of urban human settlement resilience in different periods and regions through geographical detectors.

The natural breakpoint method in ArcGIS10.2 was used to classify the detection factors and import them into the Geodetector model. The larger the $q$ value is, the greater the impact of the index on the resilience level of urban human settlements, and vice versa. Firstly, 23 indexes with $p$ values less than 0.05 were selected. Based on the research results of Pan Jinghu et al. [61], the first and second influencing factors were selected from five perspectives, namely, the natural system, human system, housing system, supporting system, and social system, and the influencing mechanism of the spatial evolution of urban human settlement resilience was discussed in depth. This study also selected the topographic relief ($X_1$), annual average temperature ($X_3$), proportion of urban population ($X_6$), population density ($X_9$), per capita green space ($X_{13}$), number of Internet connections per 10,000 people ($X_{16}$), number of buses per 10,000 people ($X_{19}$), number of library books per 100 people ($X_{27}$), per capita GDP ($X_{28}$), and per capita social consumption as 10 indicators of retail sales ($X_{33}$), which were used to analyze the main factors affecting the resilience of urban human settlements (Table 6).

**Table 6.** Detection results of 36 influencing factors of urban human settlement resilience in three provinces in Northeast China.

| Influence Factor | $q$ | $p$ | Sort |
|---|---|---|---|
| Internet per 10,000 people ($X_{16}$) | 0.7199 | 0.0000 | 1 |
| Population density ($X_9$) | 0.6693 | 0.0000 | 2 |
| Per capita GDP ($X_{28}$) | 0.6525 | 0.0000 | 3 |
| Buses for every 10,000 people ($X_{19}$) | 0.6245 | 0.0000 | 4 |
| Library volume per 100 people ($X_{27}$) | 0.6127 | 0.0000 | 5 |
| Per capita retail sales of consumer goods ($X_{33}$) | 0.6040 | 0.0000 | 6 |
| Annual mean temperature ($X_3$) | 0.5738 | 0.0000 | 7 |
| Postal service per capita ($X_{34}$) | 0.5690 | 0.0000 | 8 |
| Per capita telecommunications service ($X_{35}$) | 0.5604 | 0.0000 | 9 |
| Average wage of working staff ($X_{36}$) | 0.5337 | 0.0000 | 10 |
| Proportion of personnel in public administration and social organizations ($X_{29}$) | 0.5049 | 0.0000 | 11 |
| Per capita green space ($X_{13}$) | 0.5036 | 0.0000 | 12 |

**Table 6.** *Cont.*

| Influence Factor | q | p | Sort |
|---|---|---|---|
| Proportion of urban population ($X_6$) | 0.4811 | 0.0000 | 13 |
| Urban per capita disposable income ($X_{32}$) | 0.4738 | 0.0000 | 14 |
| Per capita domestic electricity consumption ($X_{17}$) | 0.4388 | 0.0000 | 15 |
| Per capita gas consumption ($X_{12}$) | 0.4351 | 0.0000 | 16 |
| Number of students in regular institutions of higher learning per 10,000 ($X_{15}$) | 0.3444 | 0.0000 | 17 |
| Per capita urban road area ($X_{18}$) | 0.3388 | 0.0000 | 18 |
| Per capita daily domestic water consumption ($X_{14}$) | 0.3310 | 0.0000 | 19 |
| The proportion of employees in the tertiary industry ($X_{10}$) | 0.2492 | 0.0013 | 20 |
| Gender balance rate ($X_7$) | 0.2436 | 0.0024 | 21 |
| Proportion of investment in science and technology ($X_{30}$) | 0.1015 | 0.0060 | 22 |
| Relief of relief ($X_1$) | 0.0539 | 0.0074 | 23 |
| Urban domestic sewage treatment rate ($X_{21}$) | 0.2448 | 0.3351 | 24 |
| Harmless treatment rate of household garbage ($X_{22}$) | 0.2408 | 0.3357 | 25 |
| DEM elevation ($X_2$) | 0.2255 | 0.3570 | 26 |
| The proportion of urban construction land in urban area ($X_{25}$) | 0.1678 | 0.3778 | 27 |
| Green coverage rate of built-up area ($X_{24}$) | 0.1466 | 0.4436 | 28 |
| Vegetation index ($X_5$) | 0.1073 | 0.4622 | 29 |
| Number of hospital beds per 10,000 people ($X_{20}$) | 0.1049 | 0.5513 | 30 |
| Proportion of investment in education ($X_{31}$) | 0.1015 | 0.5620 | 31 |
| Natural rate of population growth ($X_8$) | 0.0990 | 0.6072 | 32 |
| Comprehensive utilization rate of general industrial solid waste ($X_{23}$) | 0.0602 | 0.6269 | 33 |
| The proportion of people who are unemployed ($X_{11}$) | 0.0456 | 0.8930 | 34 |
| $SO_2$ emissions per unit of industrial output ($X_{26}$) | 0.0325 | 0.9549 | 35 |
| Mean annual precipitation ($X_4$) | 0.0223 | 0.9716 | 36 |

### 4.4.2. Detection of the Main Influencing Factors on the Resilience of Urban Human Settlements

In this study, the Geodetector model was used to calculate the q and *p* values of the effects of various factors on the resilience levels of urban human settlements, and the results were sorted according to the effects, as shown in Table 7. Based on the change trend in the *q* value, it can be found that there are certain differences in the degree of effect of the influencing factors on the resilience level of urban human settlements in the three provinces in Northeast China, with different time stages. From 2005 to 2020, the effects of both the "natural system" and "residential system" factors on the resilience of urban human settlements increased. (2) The influence of the "human system" and "supporting system" factors on the resilience of urban human settlements weakened. (3) The "social system" was the main factor affecting the resilience of urban human settlements, and its *q* value always ranked first. (4) The impact factors gradually changed from "society–support" to "society–residence".

**Table 7.** Detection results of the main influencing factors on the urban human settlement resilience system in three provinces in Northeast China.

| Influence Factor | 2005 | | | 2010 | | | 2015 | | | 2020 | | |
|---|---|---|---|---|---|---|---|---|---|---|---|---|
| | q | p | Sort | q | p | Sort | q | p | Sort | q | p | Sort |
| Natural system | 0.217 | 0.000 | 5 | 0.212 | 0.000 | 5 | 0.231 | 0.000 | 5 | 0.412 | 0.000 | 4 |
| Human system | 0.310 | 0.000 | 4 | 0.338 | 0.000 | 3 | 0.277 | 0.000 | 4 | 0.482 | 0.000 | 3 |
| Dwelling system | 0.426 | 0.000 | 3 | 0.457 | 0.000 | 2 | 0.609 | 0.000 | 2 | 0.505 | 0.000 | 2 |
| Support system | 0.640 | 0.000 | 2 | 0.324 | 0.000 | 4 | 0.420 | 0.000 | 3 | 0.282 | 0.000 | 5 |
| Social system | 0.788 | 0.000 | 1 | 0.729 | 0.000 | 1 | 0.645 | 0.000 | 1 | 0.563 | 0.000 | 1 |

Specifically, as shown in Table 8, the order of the effect intensity of the influencing factors on the spatial differentiation of the resilience levels of urban human settlements in 2005 is as follows: per capita retail sales of consumer goods ($X_{33}$) > number of Internet connections per 10,000 people ($X_{16}$) > buses per 10,000 people ($X_{19}$) > per capita GDP ($X_{28}$) > library collections per 10,000 people ($X_{27}$) > proportion of urban population ($X_6$) > population density ($X_9$) > terrain relief ($X_1$) > per capita green space ($X_{13}$) > annual average gas temperature ($X_3$). The main influencing factors in 2010 were the same as those in 2005: retail sales of consumer goods in human settlements ($X_{33}$), number of Internet connections per 10,000 people ($X_{16}$), buses per 10,000 people ($X_{19}$), GDP per capita ($X_{28}$) and library collections per 10,000 people ($X_{27}$). In 2015, the main influencing factors on the spatial differentiation of the resilience level of urban human settlements changed, and the factors were ranked as follows: per capita retail sales of consumer goods ($X_{33}$) > number of Internet connections per 10,000 people ($X_{16}$) > per capita GDP ($X_{28}$) > library collections per 10,000 people ($X_{27}$) > population density ($X_9$) > annual average temperature ($X_3$) > bus ownership per 10,000 people ($X_{19}$) > proportion of urban population ($X_6$) > per capita green space ($X_{13}$) > relief degree ($X_1$). In 2020, the impact factors changed significantly, and only the intensity of topographic relief ($X_1$), proportion of urban population ($X_6$), per capita green space ($X_{13}$), and per capita GDP ($X_{28}$) changed little. The main impact factors were in the following order: Internet access per 10,000 people ($X_{16}$) > population density ($X_9$) > per capita GDP ($X_{28}$) > annual average temperature ($X_3$) > library collections per 10,000 people ($X_{27}$) > per capita retail sales of consumer goods ($X_{33}$) > proportion of urban population ($X_6$) > per capita green space ($X_{13}$) > topography ($X_1$) > bus ownership per 10,000 people ($X_{19}$).

**Table 8.** Action intensity of the main influencing factors on the urban human settlement resilience system in three provinces in Northeast China.

| Influence Factor | 2005 | | | 2010 | | | 2015 | | | 2020 | | |
|---|---|---|---|---|---|---|---|---|---|---|---|---|
| | *q* | *p* | Sort | *q* | *p* | Sort | *q* | *p* | Sort | *q* | *p* | Sort |
| $X_1$ | 0.277 | 0.000 | 8 | 0.234 | 0.000 | 9 | 0.202 | 0.000 | 10 | 0.249 | 0.000 | 9 |
| $X_3$ | 0.157 | 0.000 | 10 | 0.191 | 0.000 | 10 | 0.259 | 0.000 | 6 | 0.574 | 0.000 | 4 |
| $X_6$ | 0.338 | 0.000 | 6 | 0.248 | 0.000 | 8 | 0.209 | 0.000 | 8 | 0.294 | 0.000 | 7 |
| $X_9$ | 0.282 | 0.000 | 7 | 0.429 | 0.000 | 6 | 0.349 | 0.000 | 5 | 0.669 | 0.000 | 2 |
| $X_{13}$ | 0.194 | 0.000 | 9 | 0.393 | 0.000 | 7 | 0.205 | 0.000 | 9 | 0.290 | 0.000 | 8 |
| $X_{16}$ | 0.720 | 0.000 | 2 | 0.521 | 0.000 | 5 | 0.609 | 0.000 | 2 | 0.720 | 0.000 | 1 |
| $X_{19}$ | 0.658 | 0.000 | 3 | 0.564 | 0.000 | 4 | 0.246 | 0.000 | 7 | 0.226 | 0.000 | 10 |
| $X_{27}$ | 0.629 | 0.000 | 5 | 0.643 | 0.000 | 2 | 0.593 | 0.000 | 4 | 0.474 | 0.000 | 5 |
| $X_{28}$ | 0.651 | 0.000 | 4 | 0.623 | 0.000 | 3 | 0.596 | 0.002 | 3 | 0.653 | 0.000 | 3 |
| $X_{33}$ | 0.824 | 0.000 | 1 | 0.836 | 0.000 | 1 | 0.693 | 0.000 | 1 | 0.339 | 0.000 | 6 |

Overall, there were significant changes in the influencing factors of urban human settlement resilience in the three provinces in Northeast China from 2005 to 2020. The per capita total social consumption changed from the main factor to the secondary factor, and the degree of influence of the annual average temperature and population density showed an increasing trend, gradually changing from the secondary factor to the main factor. With the passage of time, the GDP per capita ($X_{28}$), the number of Internet connections per 10,000 people ($X_{16}$), and the number of library books per 100 people ($X_{27}$) remained the main factors affecting the spatial differentiation of urban human settlement resilience in the three provinces in Northeast China.

## 5. Discussion

Because scholars have different understandings of the resilience of urban human settlements, and the establishment and selection of indicators are also different, there may be differences in the quantitative research on the resilience of urban human settlements. Li Xueming and other scholars conducted a preliminary exploration of the resilience of

urban human settlements in the Yangtze River Delta region of China, and they found that the resilience of urban human settlements varies greatly in space, and the areas with high resilience are mostly large cities or central cities [43]. In the same way, this study revealed that the distribution of urban human settlements in the three northeast provinces of China from 2005 to 2020 had significant spatial differentiation characteristics. Shenyang, Dalian, Harbin, and Daqing were the cities with high resilience, while the resilience of urban human settlements in the northern and eastern regions of the three Northeast provinces of China showed a "collapse" phenomenon. The main reason is that these regions have weak economies, imperfect infrastructure, and a single industrial structure; coupled with the "siphon effect" of large cities, this results in a continuous decline in the resilience of small- and medium-sized cities and the surrounding urban human settlements. Zhou Xiaoqi and other scholars found that the number of cities with a low resilience level among urban human settlements in China decreased year by year [45]. However, this conclusion is not valid when studying the change in the urban human settlement resilience level in the three eastern provinces from 2005 to 2020. On the one hand, COVID-19 has severely affected large, densely populated cities dominated by the tertiary sector. On the other hand, surrounding cities such as Heihe, Qiqihar, and Hegang have been less affected by COVID-19 due to their small populations. However, due to geographical restrictions and the underdeveloped transportation facilities and economy, they have always been areas with low resilience levels among urban human settlements. In addition, the difference in the results may be caused by the different research areas.

The urban human settlements in the three northeastern provinces of China usually recover, adapt, and improve after the impact of a major event. However, the resilience, recovery level, and resistance ability of the urban human settlements in the three northeastern provinces of China are also very different with different subsystems. Therefore, strategies to improve the resilience of the urban human settlements in the three northeastern provinces of China should be targeted.

(1) Adjustment path of morphological resilience. Although the $q$ value of the natural system increased from 0.217 to 0.412 during the 2005–2020 period, the effect strength of the natural system was still at a low level compared with the three subsystems including humans, habitation, and society. Therefore, it is necessary to improve the resilience of the natural system. Protecting the rich natural resources in the three eastern provinces is an important means to improve the resilience of the natural system in urban human settlements. It is necessary to return farmland to forest; increase the protection of forest land, grassland, water, and other resources; and maximize the ecological role of forest land resources. Furthermore, it is necessary to pay attention to the optimization of resources such as characteristic landscapes and green corridors, which not only facilitate the migration of organisms, but also maintain the integrity of green spaces, maintain the stability of the system, and provide rich natural resources and sufficient ecological space for urban development. At the same time, the combination of blue and green natural landscapes and urban construction land can create isolated green belts according to urban functions, which can effectively avoid the disorderly expansion of urban construction land and enhance morphological resilience.

(2) Density resilience adjustment path. Only by improving the resilience level of the human system, supporting system, and social system can the density resilience of the urban human settlement environment system be better enhanced. Specific measures are as follows.

The resilience level of the human system is in the middle level among the five subsystems of human settlements. The key to improving the resilience of the human systems is to adjust the population structure—for example, by increasing the investment in education funds to fundamentally improve the quality of the population. At the same time, the government should also implement relevant preferential policies focused on talent introduction, which will play a supporting role and help to achieve the goal of optimizing the population structure.

The role of the support system in improving the resilience of cities from 2015 to 2020 gradually diminished, mainly because of the logistics and transportation paralysis caused by COVID-19 and the absence of good policy support. Therefore, to improve the resilience of the support system, it is necessary to optimize the allocation of resources; gradually improve public resource facilities such as education, medical care, transportation, and networks, which are closely related to residents' lives, in order to realize the fairness, rationality, and accessibility of resource allocation; and provide resources for urban development. At the same time, policy reform should be carried out in education, elderly care, employment, and other aspects, as active social policies are of great significance to improving the resilience of the urban living environment.

Social systems have always played a dominant role in making cities more resilient. On the one hand, the three eastern provinces should strengthen the rational distribution of industrial space, promote the transformation and upgrading of industries with the rapid growth of the equipment manufacturing sector, exploit the potential of economic innovation, and establish a modern industrial development pattern of social sharing and industrial advancement. On the other hand, it is necessary to strengthen scientific and technological innovation; integrate digitalization, information technology, intelligence, and other advanced technologies into the transformation and construction of industries; integrate traditional industries with the economy of the new era; and use the driving force of China's large market consumption to facilitate industrial and economic progress together, so as to achieve high-quality development among urban human settlements.

(3) Regular resilience adjustment path. The central city strategy is an important tool to strengthen the resilience of urban human settlements in the three provinces in Northeast China. The Shenyang metropolitan area is not only the ninth metropolitan area in China but is also the first metropolitan area in Northeast China. In the long run, it is necessary to actively develop Anshan, Fushun, Benxi, Fuxin, Liaoyang, and other surrounding medium- and low-resilience cities; build "Shenyang" as the core; and realize the northeast region's economic integration and the regional integration of Harbin, Changchun, Shenyang, and Dalian. Among them, Shenyang's diversified industrial structure will enable in-depth cooperation with the industrial chain of the metropolitan area and even the three provinces in Northeast China, which will achieve complementary advantages. For Heihe, Chaoyang, Huludao, Shuangyashan, and other surrounding low-resilience areas, it is also necessary to make full use of the regional characteristics and continue to promote the revitalization strategy of Northeast China. Only by strengthening the radiation effect of the metropolitan areas and narrowing the differences in human settlement environment resilience caused by the large gap between central cities and marginal areas can the overall resilience level of the three provinces in Northeast China be significantly improved.

## 6. Conclusions

Based on the revitalization strategy of Northeast China, this paper discusses the spatial and temporal evolution of urban human settlement resilience and its main influencing factors in three eastern provinces from five dimensions: the natural system, human system, housing system, supporting system, and social system. Additionally, we found: (1) From the perspective of spatial and temporal pattern evolution, the average development level of urban human settlement resilience in the three provinces in Northeast China from 2005 to 2020 shows an N-shaped development trend, with significant differences among regions. The overall spatial pattern is "high in the south and low in the north", with the characteristics of "stable growth in the southwest and a gradual decline in the north". (2) Inter-regional differences are the main sources of the overall differences in the resilience of urban human settlements in the three provinces in Northeast China. The intra-regional differences in urban human settlement resilience in the three provinces in Northeast China are in the order of "Liaoning Province—Heilongjiang Province > Jilin Province—Liaoning Province > Jilin Province—Heilongjiang Province". The intra-regional differences in Liaoning Province decreased, while the intra-regional differences in Heilongjiang Province and Jilin Province

expanded. (3) At different time points, the effects of various influencing factors on the resilience level of urban human settlements in the three provinces in Northeast China are both correlated and different. From 2005 to 2020, the "social system" had the strongest influence on the resilience of urban human settlements. By combining the theory of resilience with the study of urban human settlements, this paper aimed to quantitatively analyze the resilience levels of urban human settlements in the three provinces of Northeast China and the influencing factors in order to provide a theoretical basis and practical significance for the revitalization strategy of Northeast China.

In the process of rapid urbanization, although the three provinces in Northeast China showed sustained population growth, rapid social and economic growth, increasingly adaptable service facilities, and the effective control of natural disasters, the uneven development of urban human settlement resilience remains significant. In order to improve the resilience levels of urban human settlements in the three provinces of Northeast China, corresponding measures must be taken according to the different regions and subsystems. This study not only fills the theoretical gap regarding the resilience of urban human settlements to a certain extent, but also proposes key methods and strategies to enhance the resilience of urban human settlements. Although the spatial scope of this study only included cities in the three provinces of Northeast China, it can also be used as a reference for other cities. For other regions in China, a combination of qualitative and quantitative methods can be used to quantify the resilience of urban human settlements, and, on the basis of analyzing spatial and temporal heterogeneity, the main factors affecting the resilience of urban human settlements can be explored so as to put forward reasonable suggestions.

Due to the lack of research results on the resilience of urban human settlements, there is still a great deal of work to be completed in this area. (1) Although the temporal heterogeneity in urban human settlements was analyzed in detail in this study, the time span was not long enough. In the future, dynamic research on the resilience of human settlements in long-term series should be carried out to improve its real-time performance. (2) In this study, the influencing factors of the spatial and temporal pattern evolution of the urban human settlement resilience level in the three provinces of Northeast China were only studied based on the geographical detector model, which is somewhat insufficient. The structural equation and spatial econometric model should be added in the future to further analyze the interactions among the influencing factors. (3) Although the influencing factors were identified and analyzed in this study, the number of influencing factors was limited. In the future, the indicators of the influencing factors could be enriched to improve the depth and objectivity of the resilience evaluation. In addition, the effectiveness of urban resilience can be verified in future studies by examining its performance in the face of social public health events or sudden natural disasters.

**Author Contributions:** Conceptualization, J.L. and H.L.; methodology, J.L.; software, J.L.; validation, J.L., Y.S. and X.L.; formal analysis, J.L.; investigation, J.L.; resources, J.L.; data curation, J.L.; writing—original draft preparation, J.L.; writing—review and editing, J.L.; visualization, J.L.; supervision, J.L.; project administration, X.L.; funding acquisition, X.L. All authors have read and agreed to the published version of the manuscript.

**Funding:** This research was funded by the National Natural Science Foundation of China (grant number 41671158; sponsor: Li Xueming).

**Data Availability Statement:** The data presented in this study are available upon request from the corresponding author.

**Conflicts of Interest:** The authors declare no conflict of interest.

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
