# Peer review of "The Spatial and Temporal Evolution Pattern and Influencing Factors of Urban Human Settlement Resilience in Three Provinces of Northeast China"

_land, doi:10.3390/land12122161_

Round 1

Reviewer 1 Report

Comments and Suggestions for Authors

Dear Author,

Firstly, I would like to thank you for your contribution to research the influence of the urban human settlements in Northeast China.

This is manuscript has an Systematic research methods Which is worth learning and thought. 

However, I strongly recommend you do a revision in written language, format and structure revision.

For instance:

1. Change the title.I would be more willing to read if the topic is a novel

2. The abstract does not express the paper's conclusion well or even state the contributions of the paper.(Line24-25)

3. “Dagum Gini Coefficient” is slightly insufficient as a keyword. Please update and clarify the key phrases. Please use keywords that objectively summarize all of your research methods.

4. There are structural problems in the introduction, and the logic of the research review is somewhat confused. Also, the innovation of the current research that the paper does not describe. Please rearrange this part of the framework and describe how the article is innovative or a successor from the current research

5. The description and analysis of the method are not comprehensive enough. In addition, Please supplement the steps of the research method.

6. Line 140. I'm confused about the concept of urban resilience. Please explain the relationship and difference between urban elasticity and urban living environment elasticity.

7. This paper adds two indexes(lines 148-150) to the elastic index system of urban human settlements because of the site's characteristics. Please add relevant literature to prove the relationship between the new indicators and urban environmental resilience.

8. Line 151.I can't find Table 2

9. Please make the experimental results of each part correspond to the research method of the previous part. Please redraw Figure 3 to complete the interval value information in the figure.

10. Please merge Part five(Analysis of influencing factors on resilience level of urban human settlements) with part four(Result analysis)

11. The interpretation of some data is vague. Please explain how to obtain each part of the data, calculation steps, and data support software in detail.

12. Line 233. Figure 2 is formatted incorrectly.

13. The discussion chapter is more like an introduction. Without your research progress, you also can write down the current conclusions. The previous study of various urban areas is completely irrelevant to the discussion

14. Line 487.Subheading error.

15. The conclusion does not describe the paper's contribution to the research. Besides, the future research direction in the current conclusion is more suitable for the discussion.

16. References 3 be in doubt.

17. References 34 be in doubt.

I hope the suggestions can benefit this manuscript.

Comments on the Quality of English Language

Minor check.

Author Response

Thank you for your letter and for the reviewers’ comments concerning our manuscript entitled “Spatial and temporal evolution pattern and influencing factors of urban human settlement resilience: A case study of three provinces in Northeast China” (land-2696287). Those comments are all valuable and very helpful for revising and improving our paper, as well as the important guiding significance to our researches. We have studied comments carefully and have made correction which we hope meet with approval. Revised portion are marked in red in the paper. The main corrections in the paper and the responds to the reviewer’s comments are as flowing:

Point 1: Change the title.I would be more willing to read if the topic is a novel.

Response 1: Thank you very much for your advice, This paper has changed the title to Spatial and temporal evolution pattern and influencing factors of urban human settlement resilience in three provinces of Northeast China, see lines 2-4.

Point 2: The abstract does not express the paper's conclusion well or even state the contributions of the paper.(Line24-25).

Response 2: Thank you very much for your suggestions. According to your suggestions, this paper has reorganized the abstract. The first half mainly introduces the contribution of this paper. The main purpose of studying the resilience of urban human settlements in the three northeast provinces of China is to improve the anti-risk ability of urban human settlements in the three northeast provinces of China, fully implement the strategic goal of "comprehensive revitalization of Northeast China", and realize high-quality urban development. The second half is mainly the conclusion of this paper. The conclusions of lines 24-25 are rewritten, and the conclusions are summarized from the three dimensions of overall difference, intra-regional difference and inter-regional difference respectively, and the conclusions are detailed.

Point 3: “Dagum Gini Coefficient” is slightly insufficient as a keyword. Please update and clarify the key phrases. Please use keywords that objectively summarize all of your research methods.

Response 3: Thank you very much for your suggestion, and this keyword has been deleted according to your suggestion.

Point 4: There are structural problems in the introduction, and the logic of the research review is somewhat confused. Also, the innovation of the current research that the paper does not describe. Please rearrange this part of the framework and describe how the article is innovative or a successor from the current research.

Response 4: Thank you very much for your suggestion. Now the introduction has been reorganized. The first paragraph is mainly about the research background and significance, the second paragraph is about the research progress of urban resilience at home and abroad, the third paragraph is about the research status of urban human settlement environment, the fourth paragraph is about the innovation and shortcomings of this study on urban human settlement environment, and the fifth paragraph is about the main research content of this paper.

Point 5: The description and analysis of the method are not comprehensive enough. In addition, Please supplement the steps of the research method.

Response 5: Thank you very much for your suggestions. Based on your suggestions, this paper supplements the method, adding the advantages and steps of the method in the entropy weight method and the geographical detector, and adding more detailed formula steps in the Dagum Gini coefficient decomposition method.

Point 6: Line 140. I'm confused about the concept of urban resilience. Please explain the relationship and difference between urban elasticity and urban living environment elasticity.

Response 6: Thank you very much for your suggestions. According to your suggestions, this paper has been revised accordingly. This paper believes that the relationship between urban resilience and urban human settlements resilience lies in the fact that urban human settlements resilience is subordinate to urban resilience, while regional urban resilience focuses on social indicators and mainly discusses the response of urban public service system in the face of natural disasters. The inner relation and mechanism of human settlement environment system are often neglected. The resilience of urban human settlements is more inclined to human factors, emphasizing "people" as the main body, and is manifested as a kind of anti-risk and sustainable ability that the internal structure, factors and scale of the five subsystems of human settlements gradually change from a low level to a high level when responding to internal and external shocks. See lines 204-214 for details.

Point 7: This paper adds two indexes(lines 148-150) to the elastic index system of urban human settlements because of the site's characteristics. Please add relevant literature to prove the relationship between the new indicators and urban environmental resilience.

Response 7: Thank you very much for your suggestion, the reference has now been added to the corresponding indicator, see lines 219-221.

Point 8: Line 151.I can't find Table 2.

Response 8: Thank you very much for your suggestion. I'm sorry that due to my negligence, the table name was incorrectly entered and has now been modified. Please see line 222 for details.

Point 9: Please make the experimental results of each part correspond to the research method of the previous part. Please redraw Figure 3 to complete the interval value information in the figure.

Response 9: Thank you very much for your suggestion. Now the drawing has been re-drawn according to your suggestion, as detailed in lines 384-386, and the results obtained based on which formula are added before the analysis of each part, as detailed in lines 324-326, 346, 392, 411-413 and 430-433.

Point 10: Please merge Part five(Analysis of influencing factors on resilience level of urban human settlements) with part four(Result analysis).

Response 10: Thank you very much for your suggestions, Part 4 and Part 5 have been merged according to your suggestions.

Point 11: The interpretation of some data is vague. Please explain how to obtain each part of the data, calculation steps, and data support software in detail.

Response 11: Thank you very much for your suggestion, the data source is now written in more detail, see lines 179-195.

Point 12: Line 233. Figure 2 is formatted incorrectly.

Response 12: Thank you very much for your suggestions, and Figure 2 has now been modified according to your suggestions.

Point 13: The discussion chapter is more like an introduction. Without your research progress, you also can write down the current conclusions. The previous study of various urban areas is completely irrelevant to the discussion.

Response 13: Thank you very much for your suggestion. According to your suggestion, this paper first adds the part of resilience comparison of regional urban human settlements, and also adds some research conclusions of this paper to the adjustment path proposed later. See lines 548-582, 594-599, 603-606, 613-618 for details.

Point 14: Line 487.Subheading error.

Response 14: Thanks very much for your suggestion, the title has been removed, as it was originally primarily a limitation of this article, but has been added to the conclusion.

Point 15: The conclusion does not describe the paper's contribution to the research. Besides, the future research direction in the current conclusion is more suitable for the discussion.

Response 15: Thank you very much for your suggestion. According to your suggestion, two parts are added to the conclusion, one is about the research contribution of this paper, the other is about the limitations of this paper and the direction of future research, so as to improve the academic value of this paper and better guide future academic work.

Point 16: References 3 be in doubt.

Response 16: Thank you very much for your suggestion, reference 3 has now been changed to Liu, Y.J.; Dong, F. Corruption, Economic Development and Haze Pollution: Evidence from 139 Global Countries. Sustainability 2020, 12, 3523, doi: 10.3390/su12093523.

Point 17: References 34 be in doubt.

Response 17: Thank you very much for your suggestion, reference 34 has now been changed to Doxiadis, C.A. Action for human settlements. Ekistics 1975, 40, 405-448, doi: 9787112045068.

Special thanks to you for your good comments. We tried our best to improve the manuscript and made some changes in the manuscript. These changes will not influence the content and framework of the paper.

Reviewer 2 Report

Comments and Suggestions for Authors

I have had the privilege of reviewing your scientific paper titled "Spatial and temporal evolution pattern and influencing factors of urban human settlement resilience: A case study of three provinces in Northeast China " I am pleased to provide my feedback and evaluation of your work, which I believe is highly interesting and a valuable contribution to scientific progress. The present paper is highly interesting and represents a significant contribution to scientific progress. The topic and research conducted have clear relevance and importance within the field, making it a worthwhile addition to the academic literature. Your paper is well organized, exhibiting a clear and logical structure.

I recommend the publication of this paper with some minor revisions. These revisions, I believe, will enhance the overall quality of your work and ensure that it reaches its full potential.

1) Enhance the Introduction Section (Lines 45-55). I suggest enhancing the reference in the introduction section by adding the following citation:

Tampekis, S., Sakellariou, S., Palaiologou, P., Arabatzis, G., Kantartzis, A., Malesios, C., Stergiadou, A., Fafalis, D., Tsiaras, E., 2023. "Building wildland–urban interface zone resilience through performance-based wildfire engineering. A holistic theoretical framework." Euro-Mediterranean Journal for Environmental Integration, 8, 675–689. [DOI: 10.1007/s41207-023-00385-z]

2) Caption Placement (Line 120). In Line 120, the caption of the map should be positioned below the image. Ensure that the caption's structure aligns with the formatting used throughout the manuscript.

3) References for Scholars Mentioned (3.1, Lines 132-138). When mentioning the names of scholars in Section 3.1, "Construction of resilience index system of urban human settlements," please ensure that your references adequately cover and support these citations.

4) Figure 2 Axes Titles. In Figure 2, if possible, consider reorganizing the axes titles for better clarity and presentation.

5) Conclusions Section (Lines 502-509). The content in lines 502-509 of the Conclusions section resembles a summary of the paper. To enhance the section, consider reorganizing the content to provide a more structured and concise conclusion.

6) Improvement of Discussion and Comparison with Similar Studies. The Discussion and Conclusion sections could be improved by comparing your results with similar studies if available. This comparative analysis can provide a valuable context for readers and strengthen your paper.

7) Limitations and Future Research Recommendations. In the Conclusion section, it would be valuable to specify the limitations of your research. Additionally, consider including recommendations for future research in the same area to guide future scholarly endeavors.

8) Enhancement of References. While you have provided good references, consider revisiting them to ensure that they are comprehensive and reflect the latest research in your field. I provide you some examples:

Mallick, S.K., Das, P., Maity, B., Rudra, S., Pramanik, M., Pradhan, B., Sahana, M., 2021. Understanding future urban growth, urban resilience and sustainable development of small cities using prediction-adaptation-resilience (PAR) approach. Sustainable Cities and Society 74, 103196. https://doi.org/10.1016/j.scs.2021.103196

Masnavi, M.R., Gharai, F., Hajibandeh, M., 2019. Exploring urban resilience thinking for its application in urban planning: a review of literature. International Journal of Environmental Science and Technology 16, 567–582. https://doi.org/10.1007/s13762-018-1860-2

Moradpour, N., Pourahmad, A., Hataminejad, H., Ziari, K., Sharifi, A., 2023. An overview of the state of urban resilience in Iran. IJDRBE 14, 154–184. https://doi.org/10.1108/IJDRBE-01-2022-0001

Rayan, M., Gruehn, D., Khayyam, U., 2021. Green infrastructure indicators to plan resilient urban settlements in Pakistan: Local stakeholder’s perspective. Urban Climate 38, 100899. https://doi.org/10.1016/j.uclim.2021.100899

In conclusion, your paper is a noteworthy contribution to the field, and I commend your efforts in producing a well-organized and meaningful research document. With the suggested minor revisions, I believe it will reach its full potential and become an even more valuable asset to the academic community. 

I hope you found my comments useful. Good luck and my best wishes in your further research.

Author Response

Thank you for your letter and for the reviewers’ comments concerning our manuscript entitled “Spatial and temporal evolution pattern and influencing factors of urban human settlement resilience: A case study of three provinces in Northeast China” (land-2696287). Those comments are all valuable and very helpful for revising and improving our paper, as well as the important guiding significance to our researches. We have studied comments carefully and have made correction which we hope meet with approval. Revised portion are marked in red in the paper. The main corrections in the paper and the responds to the reviewer’s comments are as flowing:

Point 1: Enhance the Introduction Section (Lines 45-55). I suggest enhancing the reference in the introduction section by adding the following citation:

Tampekis, S., Sakellariou, S., Palaiologou, P., Arabatzis, G., Kantartzis, A., Malesios, C., Stergiadou, A., Fafalis, D., Tsiaras, E., 2023. "Building wildland–urban interface zone resilience through performance-based wildfire engineering. A holistic theoretical framework." Euro-Mediterranean Journal for Environmental Integration, 8, 675–689. [DOI: 10.1007/s41207-023-00385-z]

Response 1: Thank you very much for your suggestion, the reference has been added to the article according to your comments, as shown in line 48.

Point 2: Caption Placement (Line 120). In Line 120, the caption of the map should be positioned below the image. Ensure that the caption's structure aligns with the formatting used throughout the manuscript.

Response 2: Thank you very much for your suggestion. This problem occurred because the uploaded file was not saved as PDF format, which caused the format to be confused and has now been adjusted.

Point 3: References for Scholars Mentioned (3.1, Lines 132-138). When mentioning the names of scholars in Section 3.1, "Construction of resilience index system of urban human settlements," please ensure that your references adequately cover and support these citations.

Response 3: Thank you very much for your suggestion. In response to this problem, this paper has modified the literature and added the references of Li Xueming and Peng Kunjie to ensure that this paper can fully cover and support these citations, see line 206 for details.

Point 4: Figure 2 Axes Titles. In Figure 2, if possible, consider reorganizing the axes titles for better clarity and presentation.

Response 4: Thank you very much for your suggestions, and Figure 2 has now been modified based on your suggestions.

Point 5: Conclusions Section (Lines 502-509). The content in lines 502-509 of the Conclusions section resembles a summary of the paper. To enhance the section, consider reorganizing the content to provide a more structured and concise conclusion.

Response 5: Thank you very much for your suggestion, it has been made more concise according to your suggestion, see lines 641-644.

Point 6: Improvement of Discussion and Comparison with Similar Studies. The Discussion and Conclusion sections could be improved by comparing your results with similar studies if available. This comparative analysis can provide a valuable context for readers and strengthen your paper.

Response 6: Thank you very much for your suggestion. According to your suggestion, new content is added to the discussion in this study, mainly the comparative analysis of the resilience of regional urban human settlements, aiming to compare the conclusions of this paper with the results of other papers, so as to further discuss the spatio-temporal evolution of the resilience level of urban human settlements in the three provinces of Northeast China and its causes.

Point 7: Limitations and Future Research Recommendations. In the Conclusion section, it would be valuable to specify the limitations of your research. Additionally, consider including recommendations for future research in the same area to guide future scholarly endeavors.

Response 7: Thank you very much for your suggestion. According to your suggestion, two parts are added to the conclusion, one is about the research contribution of this paper, the other is about the limitations of this paper and the direction of future research, so as to improve the academic value of this paper and better guide future academic work.

Point 8: Enhancement of References. While you have provided good references, consider revisiting them to ensure that they are comprehensive and reflect the latest research in your field. I provide you some examples:

Mallick, S.K., Das, P., Maity, B., Rudra, S., Pramanik, M., Pradhan, B., Sahana, M., 2021. Understanding future urban growth, urban resilience and sustainable development of small cities using prediction-adaptation-resilience (PAR) approach. Sustainable Cities and Society 74, 103196. https://doi.org/10.1016/j.scs.2021.103196

Masnavi, M.R., Gharai, F., Hajibandeh, M., 2019. Exploring urban resilience thinking for its application in urban planning: a review of literature. International Journal of Environmental Science and Technology 16, 567–582. https://doi.org/10.1007/s13762-018-1860-2

Moradpour, N., Pourahmad, A., Hataminejad, H., Ziari, K., Sharifi, A., 2023. An overview of the state of urban resilience in Iran. IJDRBE 14, 154–184. https://doi.org/10.1108/IJDRBE-01-2022-0001

Rayan, M., Gruehn, D., Khayyam, U., 2021. Green infrastructure indicators to plan resilient urban settlements in Pakistan: Local stakeholder’s perspective. Urban Climate 38, 100899. https://doi.org/10.1016/j.uclim.2021.100899.

Response 8: Thank you very much for your suggestion, which has been added to the article according to your suggestion, see reference 6, 23, 29, 28 for details

Special thanks to you for your good comments. We tried our best to improve the manuscript and made some changes in the manuscript. These changes will not influence the content and framework of the paper.

Reviewer 3 Report

Comments and Suggestions for Authors

The article details a resilience evaluation technique for urban areas and implements it to analyse 34 cities in China. The subject is undoubtedly relevant and of significant significance to the academic community, explored in a quantitative approach and substantiated by concrete outcomes. Potential enhancements are presented:

- The introduction could be further developed by providing a more substantial explanation of the concepts of resilience and settlement governance, potentially through the implementation of a regulatory framework or international targets. Additionally, the observed limitations of the studies already analysed should be included. 

- The arrangement of the various images and tables requires refinement: for instance, the descriptions of figures 1 and 2 are not in close proximity to the respective images they reference; thus, the numbering needs verification and all figures and tables in question should be correctly labelled within the text. 

- The formatting of table 1 should be fixed for enhanced clarity and ease of comprehension, and the target layer must be adjusted appropriately. 

- The descriptions of the tables, particularly table 1, necessitate improvement by also incorporating any used symbolism. 

- Conclusions should be expanded to encompass potential future developments and limitations that require improvement.

Author Response

Thank you for your letter and for the reviewers’ comments concerning our manuscript entitled “Spatial and temporal evolution pattern and influencing factors of urban human settlement resilience: A case study of three provinces in Northeast China” (land-2696287). Those comments are all valuable and very helpful for revising and improving our paper, as well as the important guiding significance to our researches. We have studied comments carefully and have made correction which we hope meet with approval. Revised portion are marked in red in the paper. The main corrections in the paper and the responds to the reviewer’s comments are as flowing:

Point 1: The introduction could be further developed by providing a more substantial explanation of the concepts of resilience and settlement governance, potentially through the implementation of a regulatory framework or international targets. Additionally, the observed limitations of the studies already analysed should be included. 

Response 1: Thank you very much for your suggestion. According to your suggestion, the conceptual definitions of urban resilience and urban human settlements have been added in the introduction, and corresponding national strategies have also been added, such as: In 2020, the Outline of China's 14th Five-Year Plan proposed the construction of resilient cities as a national strategy for sustainable development [30], and in order to enhance the resilience of Chinese cities, pilot projects such as adaptive cities and sponge cities were introduced [31]. In the fourth paragraph of the introduction, the shortcomings of the existing research and the innovation and contribution of this paper are emphasized.

Point 2: The arrangement of the various images and tables requires refinement: for instance, the descriptions of figures 1 and 2 are not in close proximity to the respective images they reference; thus, the numbering needs verification and all figures and tables in question should be correctly labelled within the text.

 Response 2: Thank you very much for your suggestion. This problem occurred because the uploaded file was not saved as PDF format, which caused the format to be confused and has now been adjusted.

Point 3: The formatting of table 1 should be fixed for enhanced clarity and ease of comprehension, and the target layer must be adjusted appropriately. 

 Response 3: Thank you very much for your comments. Table 1 has been modified according to your suggestions. However, since there are too many contents in Table 1 to fit on one page, only part of the contents in the table have been modified.

Point 4: The descriptions of the tables, particularly table 1, necessitate improvement by also incorporating any used symbolism. 

 Response 4: Thank you very much for your comments, and Table 1 has been modified according to your suggestions.

Point 5: Conclusions should be expanded to encompass potential future developments and limitations that require improvement.

 Response 5: Thank you very much for your suggestion. According to your suggestion, two parts are added to the conclusion, one is about the research contribution of this paper, the other is about the limitations of this paper and the direction of future research, so as to improve the academic value of this paper and better guide future academic work.

Special thanks to you for your good comments. We tried our best to improve the manuscript and made some changes in the manuscript. These changes will not influence the content and framework of the paper.

Round 2

Reviewer 1 Report

Comments and Suggestions for Authors

no further comments.

Reviewer 3 Report

Comments and Suggestions for Authors

Thanks for the improvements: now the paper sounds better. Please pay attention when referring to figures in the text: use either "Fig." or "figure" in brackets, but not both.